# Deregulated Clusterin as a Marker of Bone Fragility: New Insights into the Pathophysiology of Osteoporosis

**DOI:** 10.3390/genes13040652

**Published:** 2022-04-07

**Authors:** Virginia Veronica Visconti, Chiara Greggi, Ida Cariati, Beatrice Gasperini, Ambra Mastrogregori, Annalisa Botta, Umberto Tarantino

**Affiliations:** 1Department of Clinical Sciences and Translational Medicine, University of Rome “Tor Vergata”, 00133 Rome, Italy; virginia.veronica.visconti@uniroma2.it (V.V.V.); chiara.greggi@gmail.com (C.G.); 2Department of Orthopedics and Traumatology, PTV Foundation, 00133 Rome, Italy; ida.cariati@uniroma2.it (I.C.); beatrice.gasp95@gmail.com (B.G.); ambra.mastrogregori1993@gmail.com (A.M.); 3Department of Biomedicine and Prevention, University of Rome “Tor Vergata”, 00133 Rome, Italy; botta.med@uniroma2.it

**Keywords:** clusterin, osteoblasts, bone tissue, osteoporosis, biomarker

## Abstract

Clusterin (CLU) is a secreted heterodimeric glycoprotein expressed in all organism fluids as well as in the intracellular matrix that plays key roles in several pathological processes. Its recent involvement in muscle degeneration of osteoporotic patients led to investigation of the role of CLU in bone metabolism, given the biochemical and biomechanical crosstalk of the bone–muscle unit. Quantitative real time-polymerase chain reaction (qRT-PCR) analysis of *CLU* expression was performed in both osteoblasts and Peripheral Blood Mononuclear Cells (PBMCs) from osteoporotic patients (OP) and healthy individuals (CTR). Furthermore, immunohistochemical analysis on femoral head tissues and enzyme-linked immunosorbent assay (ELISA) in plasma samples were performed to investigate CLU expression pattern. Finally, genotyping of *CLU* rs11136000 polymorphism has also been performed by qRT-PCR assays to explore a possible association with *CLU* expression levels. Data obtained showed a significantly increased expression level of secreted *CLU* isoform in PBMCs and osteoblasts from OP patients. Immunohistochemical analysis confirms the increased expression of CLU in OP patients, both in osteocytes and osteoblasts, while plasma analysis reveals a statistically significant decrease of CLU levels. Unfortunately, no functional association between *CLU* expression levels and the presence of *CLU* rs11136000 polymorphism in OP patients was found. These data suggest a potential role played by CLU as a potential biomarker for the diagnosis and prognosis of OP progression.

## 1. Introduction

Clusterin (CLU), also known as Apolipoprotein J (ApoJ), is a highly conserved heterodimeric glycoprotein, which is expressed in a wide variety of mammalian tissues and biological fluids [1,2,3,4]. *CLU* is a single-copy gene, located on chromosome 8p21-p12 and organized into nine exons and eight introns [5]. Three main *CLU* transcripts have been identified, containing different parts of exon 1 and sharing the remaining sequence from exon 2 to exon 9 [3]. To date, two sets of different CLU isoforms have been described: a nuclear isoform (nCLU, 50–55 kDa) and a secretory isoform composed by two 40 kDa subunits resulting in a heterodimeric glycoprotein complex (sCLU, 75–80 kDa) [6]. The latter is a pro-cell survival factor involved in the regulation of cell proliferation, apoptosis, tissue remodeling, complement inhibition, lipid transport, and carcinogenesis [7,8]. Recently, CLU has been shown to play a pivotal role in age-related diseases onset and progression [9,10,11]. Genome-wide association studies (GWAS) identified significant associations between *CLU* single-nucleotide polymorphisms (SNPs) and cognitive disorders, such as late-onset Alzheimer’s disease (AD) [10,12]. Among these SNPs, several studies have focused on the role of intronic rs11136000, identifying associations between this genetic variant and the risk of AD and mild cognitive impairment (MCI) [9,10,12,13]. CLU appears also to be involved in the modulation of bone metabolism in specific aging-related diseases, including osteoarthritis (OA) and osteoporosis (OP) [7,8,14]. Higher CLU expression levels in OA cartilage [15] and also in the serum and synovial fluid of OA patients at the hip and knee were described [16]. Recently, a cytoprotective role exerted by CLU in OA disease has also been suggested, with CLU knockdown appearing to have a deleterious effect on chondrocyte proliferation, increasing the expression of inflammation and oxidative stress markers [14]. Additional data on its involvement in bone metabolism highlighted a decreased *CLU* gene expression during differentiation of mouse bone marrow-derived skeletal stem cells (mBMSCs) toward the osteoblastic lineage. The treatment of these cells with sCLU exerts an inhibitory effect on osteoblastic differentiation and an inductive effect on adipogenic differentiation, in a dose-dependent manner [8]. The role of CLU in bone metabolism was also investigated in patients with osteosarcoma, the most common malignant tumor of bone. The increased sCLU expression was positively correlated with the ability to metastasize and negatively correlated with the response to chemotherapeutic treatment, suggesting sCLU as a potential new biomarker and therapeutic target of this pathological state [17]. Interestingly, overexpression of CLU was also found in muscle atrophic and degenerated fibers of osteoporotic patients, and its silencing by siRNA led to a recovery of both myoblasts proliferative and differentiation capacity. This study identifies CLU as a degeneration marker of muscle, a target tissue of OP pathogenesis, which results in intimately connected bone constituting the bone–muscle unit, as also demonstrated for other proteins [7,18]. These data could suggest a central role of CLU in bone metabolism in different pathological conditions and direct our analysis to the characterisation of this factor in patients affected by OP, the most prevalent age-related disease of the skeletal system. In this work, we determined *sCLU* expression levels in both PBMCs and primary cultures of human osteoblasts, from a cohort of OP patients compared to control individuals. Furthermore, we investigated CLU protein expression in femoral head biopsies to validate the altered pattern in the target tissue, and in plasma samples aiming to identify a novel potential non-invasive biomarker of OP disease. Finally, we also investigated a potential functional correlation of *CLU* rs11136000 SNP and expression level of CLU in plasma samples.

## 2. Materials and Methods

### 2.1. Subjects

The study was approved by the Ethical Board of “Policlinico Tor Vergata” (approval reference number #17/21). Informed consent was obtained from all the participants and all experimental procedures were carried out according to The Code of Ethics of the World Medical Association (Declaration of Helsinki). Subjects were divided into two groups of analysis: 30 osteoporotic patients (OP) who underwent surgery for fragility fractures following low-energy trauma and 30 healthy controls (CTR) who underwent surgery for high-energy fractures. Individuals affected by malignancies, endocrine disorders affecting bone and mineral metabolism, autoimmune diseases, and bone disorders other than primary osteoporosis were excluded from the study, as well as those who underwent long-term therapy with drugs interfering with bone metabolism, sex hormone replacement therapy and/or antifracture and/or osteoanabolic therapies.

### 2.2. Clinical and Biochemical Parameters

Densitometric diagnosis of OP based on Dual-energy X-ray absorptiometry (DXA) evaluation of mineral density was carried out in each subject with a Lunar DXA apparatus (GE Healthcare, Madison, WI, USA). Lumbar spine (L1–L4) and femoral (neck and total) scans were performed according to the manufacturer’s recommendation [19]. The unit of measurement is represented by SD from the mean bone mass peak (*t*-score), and BMD was measured (in grams per square centimeter), with a coefficient of variation of 0.7%, on the uninjured limb. To further characterize the bone tissue quality of enrolled patients, histomorphometric analysis of femoral head biopsies was performed, investigating the three main quality parameters: Bone Volume (BV/TV), Trabecular Thickness (Tb.Th), and Trabecular Separation (Tb.S) (Appendix A). Finally, Calcium, PTH, and 25-(OH)-VitD levels were measured in fasting venous blood samples. The detailed clinical characteristics of the study subjects are summarized in Table 1.

### 2.3. Specimen Collection

Whole blood from participants was drawn after overnight fasting and collected in tubes with anticoagulant. An aliquot of whole blood was stored at −80 °C for DNA extraction and genotyping analysis. Furthermore, an aliquot of whole blood was transferred to a conical Falcon tube coated with Ficoll Paque. The tubes were centrifuged at 400× *g* for 20 min at 4 °C, after which the PBMCs interface was carefully removed by pipetting and washed twice with 7 mL 1× Phosphate-Buffered Saline (PBS) by centrifugation at 200× *g* for 10 min at 4 °C. PBMCs pellets were resuspended in TRIzol Reagent (Thermo Fisher Scientific, Waltham, Massachusetts, United States) and immediately frozen until further processing. Finally, to collect plasma, each blood sample was centrifuged, within an hour of collection, at 1500× *g* for 20 min at 4 °C. Plasma phase was transferred in RNase-free tubes and additional centrifugation at 16,000× *g* for 10 min at 4 °C was performed. Plasma samples were frozen in aliquots and stored at −80 °C until further processing. Femoral head biopsies (OP, *n* = 7 and CTR, *n* = 7) were collected during hip arthroplasty surgery, and subsequently fixed in 4% paraformaldehyde for 24 h and paraffin embedded after 3 days in decalcifier.

### 2.4. Human Osteoblast Primary Cell Culture

Primary osteoblast cultures were obtained from trabecular bone fragments harvested during hip replacement surgeries. These fragments were repeatedly washed in PBS and then briefly incubated at 37 °C with 1 mg/mL porcine pancreatic trypsin ≥ 60 U/mg (SERVA Electrophoresis GmbH Heidelberg, DE) diluted in PBS. After washing, bone fragments were subjected to repeated digestions with 2.5 mg/mL Collagenase NB 4G Proved grade ≥ 0.18 U/mg (SERVA Electrophoresis GmbH, Heidelberg, DE) diluted in PBS with calcium and magnesium. At the end of this step, the supernatant was collected and centrifuged at 310 RCF for 5 min. The cell pellet was resuspended in Dulbecco’s modified Eagle medium (DMEM) with 15% fetal bovine serum (FBS), seeded in a 24-well plate, and incubated at 37 °C 5% CO_2_ until confluence was reached (approximately 4 weeks). The culture medium was changed twice a week. Human osteoblasts have been characterized through the analysis of RUNX2 expression by immunocytochemical technique and *Osteocalcin* (*OCN*) expression by qRT-PCR (Appendix A).

### 2.5. RNA Extraction and qRT-PCR Analysis of Secreted Clusterin Expression

Total RNA extraction was performed by manual extraction. Chloroform totaling 0.2 mL was added to 1 mL of TRIzol reagent (Thermo Fisher Scientific, Waltham, Massachusetts, United States), shaken vigorously for 15 s, and incubated at room temperature for 3 min. Samples were centrifuged at 12,000× *g* for 15 min at 4 °C. After collection, the upper phase was mixed with isopropyl alcohol and centrifuged at 12,000× *g* for 10 min at 4 °C. The sediment was washed with 75% ethanol and air-dried for approximately 30 min. Purified RNA was dissolved in 30 μL of RNase-free water and stored at −80 °C. About 500 ng of RNA was purified and subjected to reverse transcription using the cDNA with High-Capacity Reverse Transcription kit (Thermo Fisher Scientific, Waltham, USA). Real-time PCR was performed on an Applied Biosystems^®^ 7500 Fast Real-Time PCR System (Life Technologies; Carlsbad, CA, USA). qPCR analysis was conducted using a Power SYBR green kit (Thermo Fisher Scientific, Waltham, MA, USA) and the following cycles: 95 °C for 10 min, followed by 95 °C for 15 s and 58 °C for 1 min for 40 cycles. The sequences of primers are reported in Table 2. The relative difference of *CLU* and *OCN* gene expression between OP and CTR subjects was calculated using 2^−ΔΔCT^ method and normalized to *GAPDH*, *β2-microglobulin (B2M)* and *β-actin* levels as the internal control.

### 2.6. Immunohistochemical Analysis

Serial sections of femoral head 3 μm thick were cut from formalin-fixed and paraffin-embedded specimens. Subsequently, sections were incubated with rabbit polyclonal anti-Clusterin antibody for 60 min (ab69644, AbCam, Cambridge, UK). Washing was performed with PBS/Tween20 pH 7.6 (UCS Diagnostic, Rome, Italy); reactions were revealed by horseradish peroxidase (HRP)-3,3′ diaminobenzidine (DAB) Detection Kit (UCS Diagnostic, Rome, Italy). Immunohistochemical positivity was assessed on digital images acquired with NIS-Elements software (5.30.01; Laboratory Imaging, Prague, Czech Republic): for each section, ten 20× magnification fields were analyzed and the percentage of osteoblasts and osteocytes positive for CLU expression was calculated. Each observation was performed by two researchers, with an interobserver reproducibility > 95%.

### 2.7. Quantitative Measurement of CLU in Human Plasma

Plasma CLU concentration was determined using an enzyme-linked immunosorbent assay kit (ab174447 Human Clusterin ELISA Kit, Abcam, Cambridge, UK), according to the manufacturer’s protocol. Plasma was collected using EDTA and stored frozen at −80 °C. CLU levels were all measured at the same time using reagents with the same lot numbers to reduce the measurement variability. The standard curve of the assay was constructed by applying a serial dilution. The starting concentration of CLU was set to 100,000 pg/mL and from that point diluted seven times to a final concentration of 234.4 pg/mL. The concentrations of CLU were measured in duplicates, interpolated from the CLU standard curve, and corrected for sample dilution. The absorbance was determined using Spark^®^ spectrophotometer (Tecan Trading AG, Männedorf, Switzerland), with 450 nm as wavelength.

### 2.8. Genotyping Analysis of rs11136000

Genomic DNA was extracted from the peripheral blood of each subject using FlexiGene DNA kit (Qiagen, Hilden, Germany), following the manufacturer’s instructions. Genotyping of rs11136000 SNP was determined by TaqMan assay (C_11227737_10) on Real-Time 7500 Fast PCR System (Applied Biosystem). A total of 20 ng of genomic DNA was amplified by the following PCR conditions: 95 °C for 10 min, 40 cycles of denaturation at 95 °C for 15 s, annealing and extension at 60 °C for 1 min. A final elongation was carried out at 72 °C for 10 min. Genotyping data obtained were analyzed by Cloud Dashboard application (Thermo Fisher Scientific, Waltham, MA, USA). Each run was performed including positive controls (one wildtype genotype sample, one heterozygous genotype sample and one homozygous variant genotype sample).

### 2.9. Statistical Analyses

Data were analyzed with GraphPad Prism 5.0 (GraphPad Software, Inc., La Jolla, CA, USA). Before using statistical test procedures, the assumptions of normality were verified for each variable, applying the D’Agostino and Pearson normality test. A non-parametric Mann–Whitney U-test was used for variables showing a skewed distribution whereas data following a normal distribution were processed with Welch’s test. Differences were considered significant when the *p* value was < 0.05 (* *p* < 0.05, ** *p* < 0.01, *** *p* < 0.001, **** *p* < 0.0001).

## 3. Results

### 3.1. Clinical Characteristics of Individuals Included in the Study

The baseline characteristics of the study population (OP: *n* = 30; CTR: *n* = 30) are reported in Table 1. OP patients and CTR subjects differ in terms of median age (63 ± 15.3 vs. 45.3 ± 17.8). Assessment of bone mineral density of the lumbar spine, total femur, and femoral neck, expressed as BMD and *t*-score values, showed a statistically significant difference between the OP and CTR individuals. OP and CTR subjects were characterized by a *t*-score at lumbar section of −1.0 ± 1.2 vs. 0.3 ± 1.0 (** *p* < 0.01), a *t*-score at total femur of −1.4 ± 1.1 vs. 1.0 ± 0.8 (**** *p* < 0.0001), and a *t*-score at femoral neck of −1.9 ± 0.6 vs. 0.2 ± 0.7 (**** *p* < 0.0001). In addition, there was no statistically significant difference in circulating markers between OP and CTR subjects, except for mean 25-(OH)-Vit D (ng/mL) which was lower in OP patients compared to CTR (16.2 ± 11 vs. 23.5 ± 4.5, *p* < 0.05).

### 3.2. Secreted Clusterin Expression Level Is Significantly Upregulated in Osteoblasts and PBMCs from OP Patients

In order to better explore the role played by *CLU* in bone metabolism, a gene expression analysis was carried out in the primary osteoblasts from bone, which is the target tissue mainly impaired in OP. *CLU* gene expression levels were assessed by qRT-PCR in osteoblastic cells from seven OP patients (one males and six females) and seven CTR (three males and four females). Our pilot analysis demonstrated an increased expression level of *CLU* transcript in OP patients compared to healthy subjects (Figure 1A; *p* = 0.001). Furthermore, *CLU* gene expression levels were analyzed in PBMCs samples from 30 OP patients and 30 CTRs. Our results demonstrated that the mean levels of circulating *CLU* in OP patients was higher than that of CTR individuals (Figure 1B; *p* = 0.045).

### 3.3. Immunohistochemical Analysis

Immunohistochemical analysis was conducted by a quantitative method for the evaluation of osteocytes and a semiquantitative method for the evaluation of osteoblasts. The percentage of osteocytes positive for CLU signal was calculated by making a ratio of the number of positive osteocytes to the total number of osteocytes, in 10 fields evaluated at 20× magnification. Positive osteoblasts were evaluated semi-quantitatively, from a negative (0) to a strong (+++) degree of intensity with an evaluation conducted over 10 fields at 20× magnification. In osteocytes, the percentage of signal-positive cells is higher in the bone tissue from OP patients (42.2%) than in the group of CTR patients (19%) (*p* < 0.01). In addition, osteocytes from the experimental group of OP patients show strong nuclear positivity to CLU, not detected in osteocytes of CTR individuals (Figure 2A–C). The signal intensity for CLU was again significantly higher in bone tissue osteoblasts from OP patients compared to CTR subjects (*p* < 0.001) (Figure 2D–F).

### 3.4. Clusterin Expression Level Is Significantly Downregulated in Plasma from OP Patients

CLU plasma levels in OP and CTR subjects are shown in Figure 3. Data analysis reveals a statistically significant decrease in circulating CLU levels in OP patients, compared to CTR subjects (** *p* < 0. 01). In detail, CLU plasma levels in OP patients range from 1.81 to 4.90 ng/mL, with a mean ± SD of 3.15 ± 0.87 ng/mL and a median value of 3.08 ng/mL. In the CTR subjects, CLU ranged from 2.61 to 10.30 ng/mL, with a mean ± SD of 4.99 ± 2.07 ng/mL and a median value of 4.74 ng/mL.

### 3.5. The rs1136000 Polymorphism Is Not Associated with the Expression Levels of CLU

The rs11136000 polymorphism is located within the third intron of *CLU* gene and has been associated with changes in plasma CLU levels [20]. On this basis, we decided to investigate its possible modulatory role on *CLU* expression levels also in bone diseases. Allele and genotype distributions of rs1136000 SNP in OP patients and CTRs were summarized in Table 3. The rs1136000 allelic frequencies were C = 63.3% and T = 36.7% either in OP or CTR groups. These allelic frequencies agreed with the allelic frequencies of the rs1136000 polymorphism reported in the 1.000 Genomes Project database.

In order to test if the rs1136000 polymorphism could have a functional effect on the expression level of CLU, we further analyzed the association between plasma CLU levels in OP patients and CTRs with different genotypes (Appendix A). The distribution of rs1136000 genotypes was as follows: homozygotes C/C (*n* = 10), heterozygotes C/T (*n* = 18), and homozygotes T/T (*n* = 2) in OP patients; homozygotes C/C (*n* = 11), heterozygotes C/T (*n* = 15), and homozygotes T/T (*n* = 4) in CTRs. No significant differences between OP and CTRs in the rs1136000 genotypic (*p* = 0.162) and allelic frequencies were found (*p* = 1). Deviations from Hardy–Weinberg equilibrium were not observed for rs1136000 in OP patients, indicating that this polymorphism is not associated with susceptibility to OP in the analyzed patients. No difference in circulating CLU expression levels in individuals with C/C, C/T, and T/T genotypes was detected, both in OP (C/C vs. C/T *p* = 0.192; C/C vs. T/T *p* = 0.373; C/T vs. T/T, *p* = 0.983) and CTRs (C/C vs. C/T *p* = 0.753; C/C vs. T/T *p* = 0.452; C/T vs. T/T, *p* = 0.254).

## 4. Discussion

This study identifies an altered expression pattern of CLU in different biological samples from OP patients, suggesting a novel role played by this molecule in OP disease. Interestingly, recent studies investigated the involvement of CLU in the modulation of bone and muscle metabolism, and the variable role of this protein based on the cellular microenvironment in different pathological contexts [7,14]. Our study further extends these findings, firstly revealing an increased *CLU* expression in primary osteoblast cultures of OP patients compared to healthy subjects. These data led to investigation of the expression of total CLU in bone tissue, confirming an increase in protein expression levels both in osteoblasts and osteocytes from OP patients. In addition to the main osteogenic role played by osteoblasts, it is important to analyze osteocyte population, which represent the main mechanoreceptors in bone tissue, contributing to the deposition or remodeling processes of bone [21]. Interestingly, increased CLU expression levels were also identified in human articular cartilage in vivo and in cartilage-derived chondrocytes in vitro from OA patients, suggesting in this case a possible CLU cytoprotective function [14]. Our results and those previously reported suggest how osteoarthritic and osteoporotic diseases are closely interrelated, although they affect bone tissue differently, one mainly associated with loss of trabecular bone and the other with erosion of articular cartilage. Furthermore, our results are in line with previously reported data in muscle tissue from OP patients. Indeed, both OP muscle tissue and isolated myoblasts are characterized by sCLU overexpression. Particularly, degenerated and atrophic muscle fibers highlighted a strong CLU expression, suggesting how this protein may represent a marker of muscle degeneration in these patients [7]. In addition, the treatment with exogenous CLU of myoblasts isolated from the same tissues, led to a decrease in the proliferative activity of these cells and an increase in Myogenin expression levels, the main marker of myogenic differentiation. Since the main OP target tissues set up the bone-muscle unit, these data could explain the presence of high amounts of CLU in bone tissue [22]. This altered pattern, resulting from an increase in gene expression or from an intracellular retention of the secreted form, may compromise the functionality and/or differentiation of the bone cellular component. It is noteworthy that numerous studies reported a correlation between systemic CLU levels and musculoskeletal disorders. Indeed, serum CLU levels were found to be significantly lower in patients with OA of the hand, especially in those affected by the erosive type of the disease, compared to healthy subjects. This study also reports that CLU levels have a negative association with hand pain [23]. Based on these data, we decided to assess circulating CLU levels, both by PBMCs gene expression and plasma protein expression levels analysis in OP patients compared to healthy subjects. Our analysis showed an apparently contrasting result highlighting a parallel significant upregulation in PBMCs and a lower average level of plasma CLU. Interestingly, upregulated *CLU* expression is associated with stress conditions, including oxidative stress, that characterize osteoporotic disease [24,25]. In this context, CLU can be diverted from its normal secretory pathway and be retained within the cell, which would explain the decreased plasma levels of sCLU [26]. We also studied the association between *CLU* rs11136000 intronic SNP and CLU circulating expression levels. Unfortunately, this analysis did not detect a significant association, suggesting the possibility of other SNPs modulating *CLU* expression. This manuscript includes some limitations, such as the small number of individuals analyzed, which implies the necessity for further studies on larger cohorts. In addition, a possible bias in the selection of individuals may be represented by the lower mean age of healthy subjects compared to OP patients. However, this bias is not easily resolved due to the difficulty of recruiting older individuals not affected by age-related bone diseases or other comorbidities. Further studies will be necessary to better interpret obtained data, and to clarify how CLU acts as a modulator of bone metabolism in OP pathogenesis and progression. An aging population is typically accompanied by an increased burden of several diseases, including musculoskeletal system disorders, leading to increased pressure on the global health care system [27,28]. Among these, OP imposes a high burden on the elderly population in terms of health, decreased quality of life and cost of care [29]. In this scenario our study can help the identification of novel non-invasive diagnostic markers and therapeutic targets [30], with the aim of implementing the timing of diagnosis and the treatment of OP pathology.

## Figures and Tables

**Figure 1 genes-13-00652-f001:**
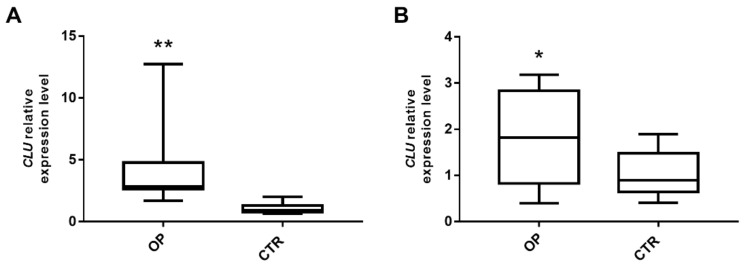
Expression of *sCLU* isoform from OP patients and CTR subjects. (**A**) *CLU* expression level was analyzed in osteoblasts from 7 OP patients and 7 CTR subjects. OP patients show an increased expression level of *CLU* with respect to CTR (** *p* < 0.01). (**B**) *CLU* expression level was analyzed in PBMCs from 30 OP and 30 CTR. OP samples show an increased expression level of *CLU* with respect to CTR (* *p* < 0.05). *GAPDH* mRNA level was used to normalize the relative amount of *CLU* and relative expression values are expressed as 2^−^^ΔΔCT^.

**Figure 2 genes-13-00652-f002:**
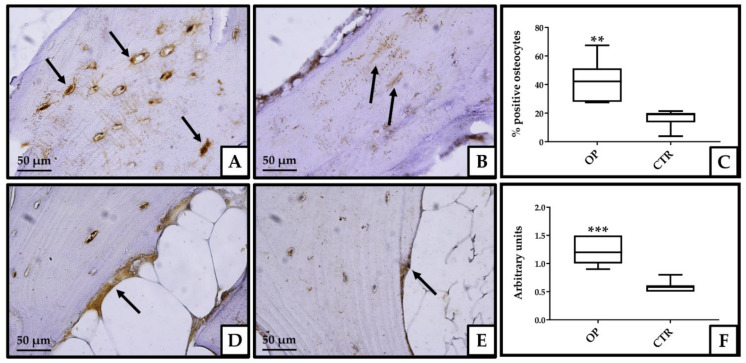
Immunohistochemical analysis of CLU protein expression in bone head biopsies. Image shows the strongly positive signal in osteocytes from the bone tissue of OP patients (**A**), whereas in the bone tissue from CTR subjects, the osteocytes were characterized by lower levels of CLU expression (**B**). The percentage of positive osteocytes was significantly higher in the experimental group of OP patients than in the CTR group (** *p* < 0.01) (**C**). Bone tissue osteoblasts from OP patients showed strong positivity to CLU signal (**D**), compared with the same bone tissue cells from CTR subjects, which were characterized by weak expression of the protein (**E**). The difference in expression intensity was evaluated semi-quantitatively and was statistically significant between OP and CTR groups (*** *p* < 0.001) (**F**). Black arrows indicate osteocytes (**A**,**B**) and osteoblasts (**D**,**E**), respectively. Images at 20× magnification, scale bar 50 μm.

**Figure 3 genes-13-00652-f003:**
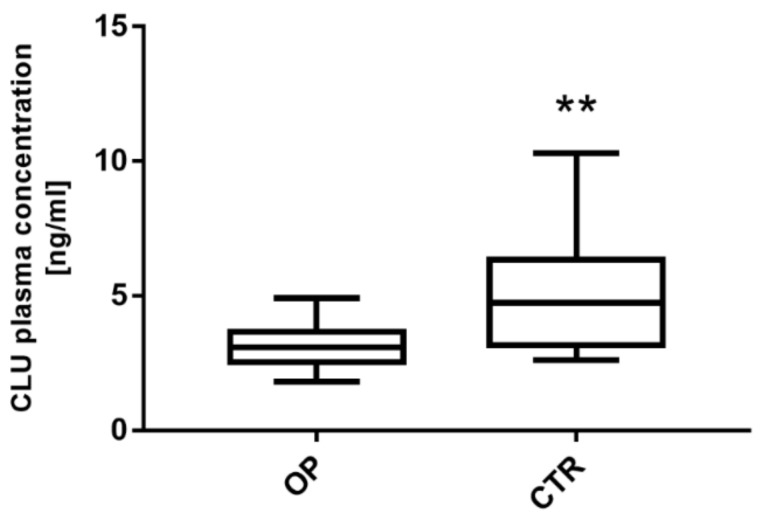
CLU plasma levels in OP patients and CTR subjects. OP patients showed a significantly lower mean circulating CLU level than CTR subjects (3.15 ± 0.87 ng/mL, 4.99 ± 2.07 ng/mL) (** *p* < 0.01).

**Table 1 genes-13-00652-t001:** Clinical characteristics of OP patients and CTR subjects.

Characteristics	OP (*n* = 30)	CTRs (*n* = 30)	*p* Value
Age (years)	63 ± 15.3	45.3 ± 17.8	** (*p* < 0.01)
BMI (Kg/cm^2^)	24.6 ± 4.5	25.8 ± 2.3	NS (*p* = 0.494)
*t*-score L1–L4	−1.0 ± 1.2	0.3 ± 1.0	** (*p* < 0.01)
*t*-score total femur	−1.4 ± 1.1	1.0 ± 0.8	**** (*p* < 0.0001)
*t*-score femoral neck	−1.9 ± 0.6	0.2 ± 0.7	**** (*p* < 0.0001)
BMD L1-L4 (g/cm^2^)	1.0 ± 0.1	1.2 ± 0.1	** (*p* < 0.01)
BMD total femur (g/cm^2^)	0.8 ± 0.1	1.2 ± 0.1	**** (*p* < 0.0001)
BMD femoral neck (g/cm^2^)	0.7 ± 0.1	1.1 ± 0.1	*** (*p* < 0.001)
Calcium (mg/dL)	8.3 ± 0.4	8.4 ± 0.3	NS (*p* = 0.614)
PTH (pg/mL)	98.9 ± 70	73.3 ± 18.5	NS (*p* = 0.410)
25-(OH)-Vit D (ng/mL)	16.2 ± 11	23.5 ± 4.5	* (*p* < 0.05)

BMI, body mass index; BMD, bone mineral density, PTH, parathyroid hormone; 25-(OH)-Vit D, 25-hydroxyvitamin D. * (*p* < 0.05); ** (*p* < 0.01); *** (*p* < 0.001); **** (*p* < 0.0001).

**Table 2 genes-13-00652-t002:** qRT-PCR primer sequences.

Gene		Sequence (5′–3′)
*CLU*	Forward	ACAGGGTGCCGCTGACC
	Reverse	CAGCAGAGTCTTCATCATGC
*OCN*	Forward	ACACTCCTCGCCCTATTG
	Reverse	GATGTGGTCAGCCAACTC
*GAPDH*	Forward	GATCATCAGCAATGCCTCCTG
	Reverse	GTCTTCTGGGTGGCAGTGAT
*B2M*	Forward	CTGGAACGGTGAAGGTGACA
	Reverse	AAGGGACTTCCTGTAACAATGCA
*β-actin*	Forward	ACTCCATGCCCAGGAAGGAA
	Reverse	GAGATGGCCACGGCTGCTT

**Table 3 genes-13-00652-t003:** Genotype distributions and allele frequencies of rs11136000 SNP in OP and CTR subjects.

	Genotype			*p*	Allele		*p*	Odds Ratio (95% CI)
	C/C	C/T	T/T		C	T		
OP (*n* = 30)	10 (33.3%)	18 (60%)	2 (6.7%)	0.162	38 (63.3%)	22 (36.7%)	1	1 (0.563–1.777)
CTRs (*n* = 30)	11(37%)	15 (50%)	4 (13%)		38 (63.3%)	22 (36.7%)		

## Data Availability

The data presented in this study are available on request from the corresponding author.

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
