# Peer review of "Deregulated Clusterin as a Marker of Bone Fragility: New Insights into the Pathophysiology of Osteoporosis"

_genes, 2022, doi:10.3390/genes13040652_

Round 1
Reviewer 1 Report
Visconti et al. have revised their original submission and added new results, as suggested during earlier round of revision. Terminology for “serum CLU expression” has been changed to expression analysis on PBMCs, methods have been altered accordingly (from miRNeasy kit to total RNA isolation by Trizol). The manuscript has thus improved, with better description of methods (DXA, RNA analysis, new ELISA assays), improved layout of figures (boxplots) and more relevant data. The discussion is also more balanced, with recognition of limitations of the study. I have a few comments:
1) Methods are better described, but the validation of primary human osteoblast culture is still rather weak, based on only Runx2 IHC (Fig.S1). Fig.S1 should be complemented with negative staining control (w/o prim. antibody, for DAB background). Counterstaining with haematoxylin is mentioned in figure legend but not shown in FigS1?, Since you have mRNA isolated from osteoblasts (Fig. 1A), why authors do not analyze an osteoblast marker gene (eg. Runx2, ALP, Osteocalcin…) from these samples, to verify the osteoblastic phenotype and to see if CLU expression behaves similarly/differently to osteoblastic genes?
2) Background information and the number of patients for femoral head biopsies is still missing in paragraph 2.3. Are these biopsy samples (results shown in Fig.3) identical to those samples used for in vitro osteoblast culture (n=7 OP patients and n=7 controls, paragraph 3.3.), Please, clarify the origin of biopsy samples in Methods.
3) The study cohort is not the same as in earlier submission. Earlier, 46 OP patients and 28 CTRs were used, but in the revised submission 30 OP patients and 30 CTRs are described. Why? Are these partially same subjects as before?
4) Statement “In this scenario our study can help the identification of new diagnostic markers and therapeutic targets [30–32]...” is very general, and does not require references. They could be deleted, particularly as they are all self-citations. Three references could be replaced by only one of them, or perhaps better with a review article introducing several emerging markers.
Author Response
Reviewer 1
Visconti et al. have revised their original submission and added new results, as suggested during earlier round of revision. Terminology for “serum CLU expression” has been changed to expression analysis on PBMCs, methods have been altered accordingly (from miRNeasy kit to total RNA isolation by Trizol). The manuscript has thus improved, with better description of methods (DXA, RNA analysis, new ELISA assays), improved layout of figures (boxplots) and more relevant data. The discussion is also more balanced, with recognition of limitations of the study. I have a few comments:
1) Methods are better described, but the validation of primary human osteoblast culture is still rather weak, based on only Runx2 IHC (Fig.S1). Fig.S1 should be complemented with negative staining control (w/o prim. antibody, for DAB background). Counterstaining with haematoxylin is mentioned in figure legend but not shown in FigS1? Since you have mRNA isolated from osteoblasts (Fig. 1A), why authors do not analyze an osteoblast marker gene (eg. Runx2, ALP, Osteocalcin…) from these samples, to verify the osteoblastic phenotype and to see if CLU expression behaves similarly/differently to osteoblastic genes?
Response: We thank the reviewer for the suggestion. We proceeded to repeat immunocytochemistry of RUNX2, adding negative CTR, counterstaining with haematoxylin. We also implemented osteoblast characterization by performing Osteocalcin gene expression analysis either in OP and CTR subjects, using as negative control mRNA isolated from myoblasts culture (Figure S1).
2) Background information and the number of patients for femoral head biopsies is still missing in paragraph 2.3. Are these biopsy samples (results shown in Fig.3) identical to those samples used for in vitro osteoblast culture (n=7 OP patients and n=7 controls, paragraph 3.3.), Please, clarify the origin of biopsy samples in Methods.
Response: We thank the reviewer for the helpful comment. We have included the background information and the number of patients from whom femoral head biopsies were taken in section 2.3.
3) The study cohort is not the same as in earlier submission. Earlier, 46 OP patients and 28 CTRs were used, but in the revised submission 30 OP patients and 30 CTRs are described. Why? Are these partially same subjects as before?
Response: Unfortunately, we did not have enough starting material (i.e. plasma) from the first case series (46 Vs 28) to perform the ELISA assay requested in the first revision of the manuscript. We were therefore forced to analyze samples recruited for another research project. In this new study cohort we have repeated either PBMCs and genotyping analyses and extend the characterization with ELISA assay.
4) Statement “In this scenario our study can help the identification of new diagnostic markers and therapeutic targets [30–32]...” is very general and does not require references. They could be deleted, particularly as they are all self-citations. Three references could be replaced by only one of them, or perhaps better with a review article introducing several emerging markers.
Response: We thank the reviewer for the suggestion, only one Reference has been included (ref 30).

Reviewer 2 Report
The authors performed an extensive work on the clusterin protein and its involvement in osteoporosis.The study is well done and very interesting results have been obtained, although the conclusions are not in line with the study's approach.
The main problem is the age difference between both groups (OP vs CTRL), which would suggest that differences in CLU expression between groups are due to age and not to osteoporosis.
This is also reflected in the fact that they find differences in both plasma and PBMCs, therefore it seems a systemic condition related to aging rather than an issue of OP bone fracture specifically. It should be noted that the femoral neck t-score of the OP group does not fit to the parameters described by the WHO which are <-2.5. Moreover, z-score is more useful when comparing between different age groups.
Other point is the gender differences since postmenopausal OP could have different underlying mechanisms than male OP. In addition, postmenopausal bone should not be compared to young bone.
It should be compared between bones of the same age with only difference OP vs. No OP. For example, bone specimens could be obtained from femoral heads obtained from prothesis replacement due to OA.
In summary, comparisons should be done between more homogeneous groups to conclude that the only factor is the OP. On the contrary, conclusions should go towards the ageing, or postmenopausal associations.
Minor considerations:
RUNX2 is a marker of osteoblast differentiations instead of mature osteoblast. RUNX2 defines an osteoblastic lineage. For mature osteoblasts is better to use osteocalcin or ALP.
Since clusterin expression was evaluated in the bone context, it is not clear why authors analyze genotypes with CLU levels in plasma and not in osteoblasts or bone tissue. Of note, CLU plasma levels and CLU bone levels seems to have a contrary behavior.
In section 2.6.
“for each section, ten 20× magnification fields were analyzed at and the percentage of osteoblasts and osteoclasts positive for CLU expression has been calculated.”
I think that osteoclasts are not assessed.
Figure S2. Merging both figures (A and B) is suggested to better compare between groups. SD is lacking in the TT genotype in CTR group.
Author Response
Reviewer 2
The authors performed an extensive work on the clusterin protein and its involvement in osteoporosis. The study is well done and very interesting results have been obtained, although the conclusions are not in line with the study's approach.
The main problem is the age difference between both groups (OP vs CTRL), which would suggest that differences in CLU expression between groups are due to age and not to osteoporosis.
Response: we thank the reviewer for the suggestion. However, both OP patients and CTR subjects show a high standard deviation. Some CTR subjects of the same age as OP patients were included, and vice versa, leading to the suggestion that CLU expression between the two groups is not age-related. Furthermore, unfortunately enrolling CTR subjects of the same age as OP, who do not have osteoporosis or other comorbidities is extremely difficult.
This is also reflected in the fact that they find differences in both plasma and PBMCs, therefore it seems a systemic condition related to aging rather than an issue of OP bone fracture specifically. It should be noted that the femoral neck t-score of the OP group does not fit to the parameters described by the WHO which are <-2.5. Moreover, z-score is more useful when comparing between different age groups.
Response: We thank the reviewer for his careful observation. The mean t-score value reported for both femoral neck and total femur does not correspond to -2.5, because some patients who had a t-score of -2.5 at the femoral neck did not have the same value for the total femur. In the same way as some patients who had a t-score of -2.5 at the total femur but reported a t-score value closer to zero at the femoral neck. Furthermore, applying the standard deviation to the mean value of t-score, a value of -2.5 is reached in both cases. All patients enrolled in the OP experimental group were included both considering DXA values and fracture type: all fractures to which patients in the OP group were low-energy fractures (in almost all cases, falls in the home environment). According with literature data and Ministry of Health, a fragility fracture, i.e. osteoporosis, is in fact defined as a fracture occurring as a result of a trauma that would not damage a normal bone. In addition, since our research group has been involved for many years in the study of osteoporosis and osteoarthritis, histomorphometric analysis of bone tissue is performed on all patients enrolled. Bone Volume (BV/TV), Trabecular Thickness (Tb.Th) and Trabecular Separation (Tb.S) values are in fact indicative of a fragile state of the bone tissue for OP patients, compared to CTRs who are instead characterized by histomorphometric parameters indicative of good tissue quality (Figure S3). We follow this protocol since, also according to what the literature reports, the DXA method may incur several errors of evaluation (Nelson B. Watts et al., 2004).
Other point is the gender differences since postmenopausal OP could have different underlying mechanisms than male OP. In addition, postmenopausal bone should not be compared to young bone. It should be compared between bones of the same age with only difference OP vs. No OP. For example, bone specimens could be obtained from femoral heads obtained from prothesis replacement due to OA. In summary, comparisons should be done between more homogeneous groups to conclude that the only factor is the OP. On the contrary, conclusions should go towards the ageing, or postmenopausal associations.
Response: we thank the reviewer for the comment. In our study, the femoral heads from OP patients (6 females Vs 1 male) were selected including women without post-menopausal osteoporosis in order to make the OP Vs CTRs comparison more homogeneous. In addition, we did not consider it correct to use femoral heads obtained from prothesis replacement due to OA, as several studies in the literature show that OP and OA are highly interconnected. These studies (PMID 31394795, 26780427, 31934183) describe the presence of risk factors, epigenetic mechanisms of gene expression regulation and pathogenetic processes themselves common between the two bone diseases, and we decided to exclude OA patients as surrogate controls for OP disease.
Minor considerations:
RUNX2 is a marker of osteoblast differentiations instead of mature osteoblast. RUNX2 defines an osteoblastic lineage. For mature osteoblasts is better to use osteocalcin or ALP.
Response: We thank the reviewer for the suggestion. We also implemented osteoblast characterization by performing Osteocalcin gene expression analysis either in OP and CTR subjects, using as negative control mRNA isolated from myoblasts culture (Figure S1).
Since clusterin expression was evaluated in the bone context, it is not clear why authors analyze genotypes with CLU levels in plasma and not in osteoblasts or bone tissue. Of note, CLU plasma levels and CLU bone levels seems to have a contrary behavior.
Response: We thank the reviewer for the helpful comment. The correlation between CLU levels detected in bone tissue and osteoblasts, and genotypes was not conducted due to the very small number of biopsies (n=7), which would result in uninformative data. Furthermore, literature data identify correlations between genotype and CLU expression with respect to systemic levels of the protein.
In section 2.6. “for each section, ten 20× magnification fields were analyzed at and the percentage of osteoblasts and osteoclasts positive for CLU expression has been calculated.” I think that osteoclasts are not assessed.
Response: We thank the reviewer for the helpful comment. We have corrected the error by replacing “osteoclasts” with “osteocytes”.
Figure S2. Merging both figures (A and B) is suggested to better compare between groups. SD is lacking in the TT genotype in CTR group.
Response: We thank the reviewer for the suggestion and have merged both figures (A and B). In addition, we have added SD in the T/T genotype of the CTRs group.
